# New Control Scheme for Solar Power Systems under Varying Solar Radiation and Partial Shading Conditions

Anindya-Sundar Jana [1], Hwa-Dong Liu [2], Shiue-Der Lu [3] and Chang-Hua Lin [1,*]

1   Department of Electrical Engineering, National Taiwan University of Science and Technology,
    Taipei 106, Taiwan; D10907808@mail.ntust.edu.tw
2   Undergraduate Program of Vehicle and Energy Engineering, National Taiwan Normal University,
    Taipei 106, Taiwan; D10507003@mail.ntust.edu.tw
3   Department of Electrical Engineering, National Chin-Yi University of Technology, Taichung 411, Taiwan;
    sdl@ncut.edu.tw
*   Correspondence: link@mail.ntust.edu.tw; Tel.: +886-227303289

**Abstract:** The traditional perturbation and observation (P&O) maximum power point tracking (MPPT) algorithm of a structure is simple and low-cost. However, the P&O algorithm is prone to divergence under solar radiation when the latter varies rapidly and the P&O algorithm cannot track the maximum power point (MPP) under partial shading conditions (PSCs). This study proposes an algorithm from the P&O algorithm combined with the solar radiation value detection scheme, where the solar radiation value detection is based on the solar photovoltaic (SPV) module equivalent conductance threshold control (CTC). While the proposed algorithm can immediately judge solar radiation, it also has suitable control strategies to achieve the high efficiency of MPPT especially for the rapid change in solar radiation and PSCs. In the actual test of the proposed algorithm and the P&O algorithm, the MPPT efficiency of the proposed algorithm could reach 99% under solar radiation, which varies rapidly, and under PSCs. However, in the P&O algorithm, the MPPT efficiency was 96% under solar radiation, which varies rapidly, while the MPPT efficiency was only 80% under PSCs. Furthermore, in verifying the experimental results, the proposed algorithm's performance was higher than the P&O algorithm.

**Keywords:** solar photovoltaic module; conductance threshold control; partial shading conditions

## 1. Introduction

On rainy or cloudy days, solar radiation is under 150 W/m² and the solar power generation (SPG) system output power is low [1]. The SPG system output energy is dependent on climatic elements (e.g., solar radiation and ambient temperature). Therefore, the maximum power point tracking (MPPT) controller can greatly improve the efficiency of the SPG system [2,3].

Numerous MPPT algorithms are available for the SPG system and have been extensively investigated [4–23]. The hill-climbing (HC) algorithm architecture is simple in that only two power points are compared and then MPPT is executed [4]. Meanwhile, the perturbation and observation (P&O) algorithm compares the relationship between voltage and power slope and then searches for the maximum power point (MPP) [5]. The incremental conductance (INC) method implements MPPT based on the relationship between the incremental conductance (dI/dV) and the conductance (I/V) [6]. The fuzzy logic algorithm is a computer-intelligent control based on fuzzy variables and fuzzy logic inferences [7]. The neural network algorithm is a mathematical model that imitates the structure and function of biological neural networks [8]. Further, the particle swarm optimization (PSO) algorithm is based on imitating the behavior of bird flocks and discovering the advantages of evolution in bird flocks [9]. The artificial bee colony (ABB) algorithm was developed by the bee foraging method [10] while the intelligent monkey king evolution

(IMKE) is a control strategy developed based on the super ability of the Chinese novel Monkey King [11]. The flower pollination (FP) method is an algorithm for pollen transfer in nature [12] and grey wolf optimization (GWO) is an algorithm inspired by the behavior of gray wolves [13]. The data-driven MPPT method in PV systems is based on voltage and power and searches for the MPP [14] while the state-plane direct MPPT algorithm is the large-signal and state-plane model of the converter developed [15]. The leaky least logarithmic absolute difference-based control algorithm and the learning-based INC MPPT algorithm improve INC method problems such as steady-state oscillations and slow dynamic responses [16] and evaluate the characteristics of the HC algorithm and INC method to discover the applicable area [17]. The purpose of the proposed strategy is to combine the P&O algorithm and the fireworks algorithm (FWA), which can track the MPP in partial shading conditions (PSCs) [18]. The new fuzzy logic control (FLC) technology developed based on the HC algorithm [19] discusses the different MPPT technologies and analyzes the applied conditions. As a research reference [20], the research introduces the adaptively binary weighted steps followed by the monotonically decreased step. In addition, as an MPPT technique [21], it studies the turbulent flow of water-based optimization (TFWO) to analyze the characteristics of SPV modules [22]. Furthermore, the research proposes the pigeon-inspired optimization (PIO) algorithm, a new type of metaheuristic algorithm, which implements MPPT under PSCs [23].

The P&O algorithm, whose structure is simple and low-cost, is the most frequently used [17]. However, it has four disadvantages: (1) while it cannot track the MPP under partial shading, this algorithm could converge to the local peak power point (LPPP) and cause a lower system performance [18]; (2) with the P&O algorithm's actuating point close to the maximum power point (MPP), it converges slowly [19]; (3) this algorithm's actuating point oscillates near the MPP when solar radiation is steady, causing low system efficiency [19,20]; (4) the P&O algorithm is prone to divergence when solar radiation varies quickly [20,21].

The present study proposes a new control scheme for the SPG system where the solar radiation value detection scheme is the SPV module equivalent conductance ($R_{spv}^{-1}$) threshold control (CTC) implemented in the MPPT control scheme, combined with the P&O algorithm. The experimental comparison is of the proposed algorithm and the P&O algorithm under varying solar radiation and partial shading conditions. Moreover, the proposed algorithm efficiency is higher than the P&O algorithm, which could operate at the MPP and avoid being trapped in the LPPP under PSCs.

## 2. Perturbation and Observation Algorithm

The P&O algorithm is based on the SPV module $P_{spv}$-$V_{spv}$ characteristic curve *slope* ($dP_{spv}/dV_{spv}$). If the P&O algorithm actuating point is on the left-half plane (LHP) for the SPV module $P_{spv}$-$V_{spv}$ characteristic curve, it means that the slope is positive. On the contrary, if the P&O algorithm actuating point is on the right-half plane (RHP) for the SPV module $P_{spv}$-$V_{spv}$ characteristic curve, it indicates that the slope is negative. This P&O algorithm depends on the slope and further perturbs the duty cycle to track the MPP. However, this algorithm's actuating point oscillates near the MPP, causing low system efficiency. In addition, the SPV module under partial shading implies that this algorithm's actuating point could converge to the SPV module's local maximum power point (LMPP), resulting in power loss [24].

## 3. Proposed Algorithm

When solar radiation varies rapidly, the traditional P&O algorithm's actuating point is continuously perturbed and does not immediately catch the MPP, thereby causing low system efficiency. In order to solve this problem, the proposed CTC is integrated with the P&O algorithm. This proposed algorithm can estimate the actual solar radiation value and execute the MPPT, improving the system efficiency.

Moreover, solar radiation is stable as in Equation (1). This proposed algorithm is to keep a constant duty cycle and improve the perturbation problem. Notably, this greatly improves the MPPT efficiency of the P&O algorithm.

$$slope = \frac{dP_{spv}}{dV_{spv}} = 0. \tag{1}$$

The proposed algorithm first executes the MPPT to track the MPP where the *slope* is 0, which will then be entered in the CTC mode. The illustration of the proposed algorithm control principle is as follows: (1) Figure 1 illustrates the $I_{spv}$-$V_{spv}$ characteristic curves of the SPV module, which has been informed. When solar radiation G and temperature T change, so do the SPV module output voltage $V_{spv}$ and the current $I_{spv}$ (2). The important parameter, $R_{spv}$, of Equation (2) changes based on G and T. Thus, $R_{spv}$ can reflect G and T changes and $R_{spv}$ of the SPV module is an important reference factor for the CTC.

$$P_{spv} = \frac{V_{spv}^2}{R_{spv}} = V_{spv}^2 \cdot R_{spv}^{-1}. \tag{2}$$

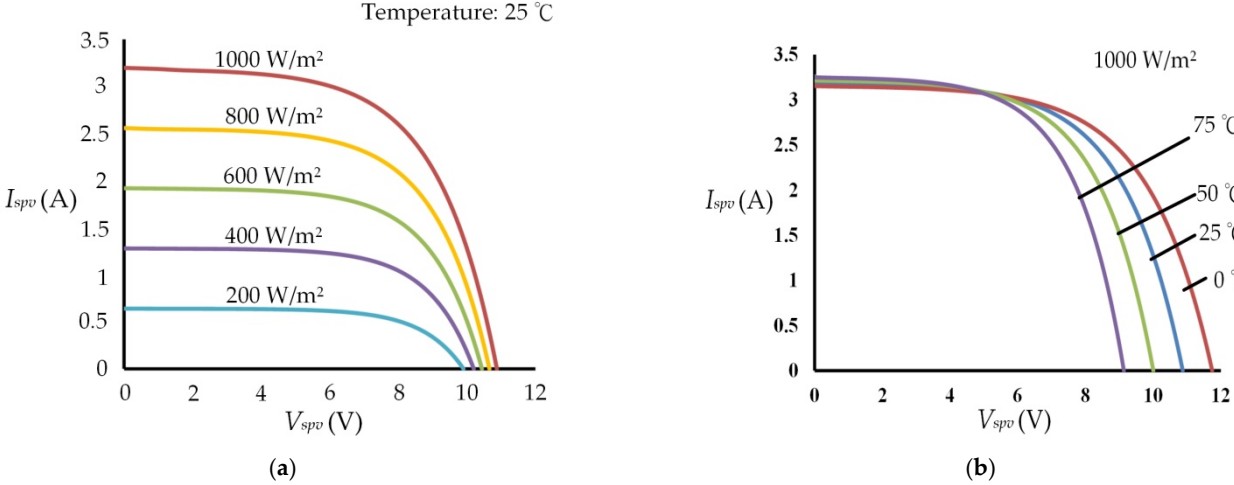

**Figure 1.** A single SPV module $I_{spv}$-$V_{spv}$ characteristic curve graph (Everbright, model number Q025). (**a**) T of 25 °C; G of 200, 400, 600, 800 and 1000 W/m², respectively. (**b**) G of 1000 W/m²; T of 0, 25, 50 and 75 °C, respectively.

The SPV module (Everbright, model number Q025) was used during the experiment. Figure 1a shows the SPV module $I_{spv}$-$V_{spv}$ characteristic curves of the T of 25 degrees and the G of 200, 400, 600, 800 and 1000 W/m². Figure 1b shows the SPV module $I_{spv}$-$V_{spv}$ characteristic curves of the G of 1000 W/m² and the T of 0, 25, 50 and 75 °C.

In this study, the proposed method (based on Figure 1a,b $I_{spv}$-$V_{spv}$ characteristic curves) converted the relationship between $I_{spv}$ and $R_{spv}^{-1}$ through Microsoft Excel and presented it with trend lines. Hence, four trend lines were drawn (as in Figure 2) to illustrate the relationship between $I_{spv}$ and $R_{spv}^{-1}$ as follows: line A for the T of 0 °C and the G of 0–1000 W/m², line B for the T of 25 °C and the G of 0–1000 W/m², line C for the T of 50 °C and the G of 0–1000 W/m² and line D for the T of 75 °C and the G of 0–1000 W/m². The mathematical model of the four trend lines could be approximated by the following quadratic equation, simplified as Equation (3).

$$R_{spv}^{-1} = a\left(I_{spv}\right)^2 + b\left(I_{spv}\right) + c. \tag{3}$$

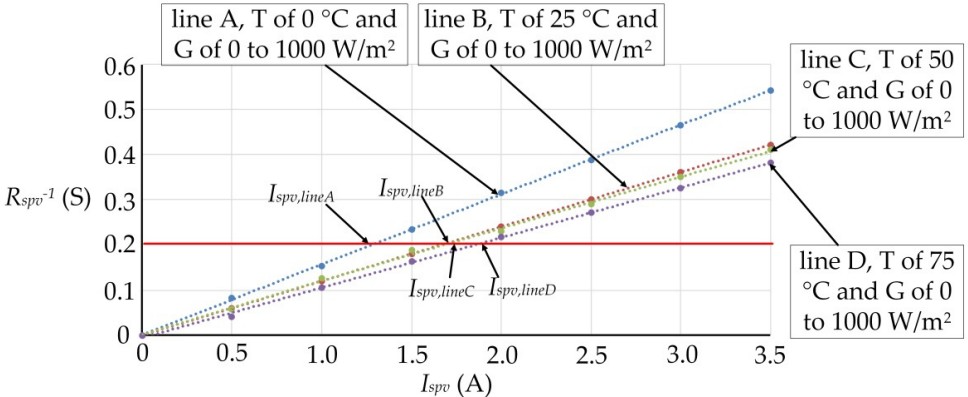

**Figure 2.** Relationship between $R_{spv}^{-1}$ and $I_{spv}$.

$I_{spv}$ is obtained by Equation (3) as follows:

$$I_{spv} = \frac{-b + \sqrt{b^2 - 4a\left(c - R_{spv}^{-1}\right)}}{2a}. \tag{4}$$

In Figure 2, line A was drawn with Equation (3) where the parameters $a$, $b$ and $c$ are $-0.0007$, $0.1572$ and $-0.0005$, respectively. Line B was drawn with the parameters $a$, $b$ and $c$ as $0.0002$, $0.1197$ and $0.00003$, respectively. Line C was drawn with the parameters $a$, $b$ and $c$ as $-0.0013$, $0.1204$ and $-0.0012$, respectively. Line D was drawn with the parameters $a$, $b$ and $c$ as $-0.0001$, $0.1112$ and $-0.0058$, respectively.

When the G and T change, the corresponding points of $R_{spv}^{-1}$ and $I_{spv}$ range from line A to line D, as shown in Figure 2. In this study, $R_{spv}^{-1} = P_{spv}/V_{spv}^2$, according to Equation (4), to calculate $I_{spv,line}$ (e.g., $I_{spv,lineA}$, $I_{spv,lineB}$, $I_{spv,lineC}$ and $I_{spv,lineD}$). As shown in Figure 2, when $R_{spv}^{-1} = 0.2$ S, $I_{spv,lineA}$, $I_{spv,lineB}$, $I_{spv,lineC}$ and $I_{spv,lineD}$ are different. Although the four trend lines have the same $R_{spv}^{-1}$, a different T and G draw different trend lines and the calculated $I_{spv,line}$ will be significantly different.

In this study, the proposed method, based on Figure 1a,b $I_{spv}$-$V_{spv}$ characteristic curves, converted the relationship between $R_{spv}^{-1}$ and the G through Microsoft Excel and presented it with trend lines. The four trend lines were drawn (as in Figure 3) to show the relationship between the $R_{spv}^{-1}$ and the G as follows: line A.1 for the T of 0 °C and the G of 0–1000 W/m², line B.1 for the T of 25 °C and the G of 0–1000 W/m², line C.1 for the T of 50 °C and the G of 0–1000 W/m² and line D.1 for the T of 75 °C and the G of 0–1000 W/m². The mathematical model of the four trend lines could be approximated by the following quadratic equation, simplified as Equation (5):

$$G = d\left(R_{spv}^{-1}\right)^2 + e\left(R_{spv}^{-1}\right) + f. \tag{5}$$

In Figure 3, line A.1 was drawn with Equation (5) using $d = 9 \times 10^{-11}$, $e = 2612.8$ and $f = 2 \times 10^{-12}$; line B.1 was drawn with $d = 662.46$, $e = 3148.2$ and $f = -0.2585$; line C.1 was drawn with $d = -232.85$, $e = 3541.1$ and $f = -5.7651$ and line D.1 was drawn with $d = 6 \times 10^{-11}$, $e = 3575.5$ and $f = -5 \times 10^{-13}$.

Figures 2 and 3 have a corresponding relationship between each other. If the values of $R_{spv}^{-1}$ and $I_{spv}$ fall on line A in Figure 2, they correspond with line A.1 in Figure 3 and then the G can be calculated by Equation (5). Furthermore, if the values of $R_{spv}^{-1}$ and $I_{spv}$ fall on line B or C or D in Figure 2, they correspond with line B.1 or C.1 or D.1 in Figure 3, respectively, and then the G can be calculated by Equation (5).

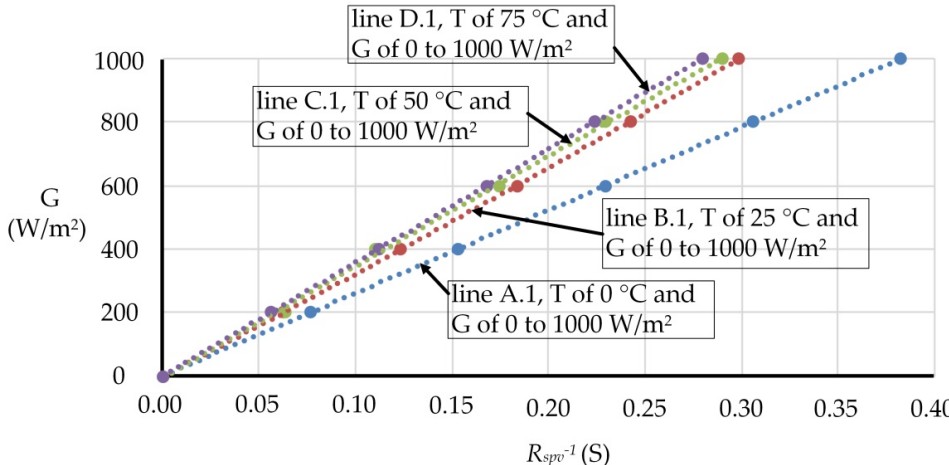

**Figure 3.** Relationship between $R_{spv}^{-1}$ and the irradiance level.

The proposed algorithm can calculate the G to improve the MPPT range and accuracy. Equation (5) is derived from Equations (2)–(4) and Figure 1a,b. Therefore, the proposed algorithm under a different G and T can accurately track the MPP and improve the MPPT performance.

A comparison of the four trend lines in Figure 3 shows that (1) the mean deviations of line A and line B were 6%, the mean deviations of line B and line C were 0.5% and the mean deviations of line C and line D were 1.8%; (2) assuming $I_{spv} > 1.06 \cdot I_{pv,lineB}$, it falls in the interval of line A; (3) if $I_{spv} \leqq 0.94 \cdot I_{spv,lineA}$ or $I_{spv} > 1.005 \cdot I_{spv,lineC}$, it falls in the interval of line B; (4) assuming $I_{spv} \leqq 0.995 \cdot I_{spv,lineB}$ or $I_{spv} > 1.018 \cdot I_{spv,lineD}$, it falls in the interval of line C; (5) assuming $I_{spv} \leqq 0.982 \cdot I_{spv,lineC}$, it falls in the interval of line D.

In order to determine the sudden change of G, the proposed algorithm used a continuous detection G variation value (dG). Generally, a sudden change in G is small in magnitude (smaller than 27 W/m$^2$) [25]. In this control scheme judgment, G does not change, which reduces unnecessary vibrations of the actuating point. Therefore, in this study, the CTC threshold value was set to 27 W/m$^2$. Once the G change was detected to be more than 27 W/m$^2$, the proposed algorithm tracked the new MPP.

The designer's actual demand sets the value of the CTC threshold. If the value of the CTC threshold is too small, the response is fast and the actuating point oscillations around the MPP cause power loss. On the contrary, when the value of the CTC threshold is too large, the response is slow and the MPPT will lack precision.

Figure 4a illustrates that when time = 0 s, there is a G of 600 W/m$^2$, $P_{spv}$ = 12 W and a duty cycle of 0.7, then when time = 0.2 s, the G of 600 W/m$^2$ drops to 500 W/m$^2$. Thus, the G variation value is more than 27 W/m$^2$. Therefore, the proposed algorithm starts to track the new MPP. The $P_{spv}$ of 10 W and duty cycle of 0.6 are shown in Figure 4b,c. Figure 4a displays that when time = 0.4 s, the G of 500 W/m$^2$ drops down to 490 W/m$^2$. Thus, the G variation value is less than 27 W/m$^2$. Therefore, the proposed algorithm to calculate the duty cycle is fixed, preventing perturbations that cause power loss. The $P_{spv}$ of 9.8 W and the duty cycle of 0.6 are shown in Figure 4b,c. Figure 4a illustrates that when time = 0.6 s, the G increases from 490 W/m$^2$ to 500 W/m$^2$. Thus, the G variation value is less than 27 W/m$^2$ and that the duty cycle is also fixed. The $P_{spv}$ of 10 W and duty cycle of 0.6 are shown in Figure 4b,c.

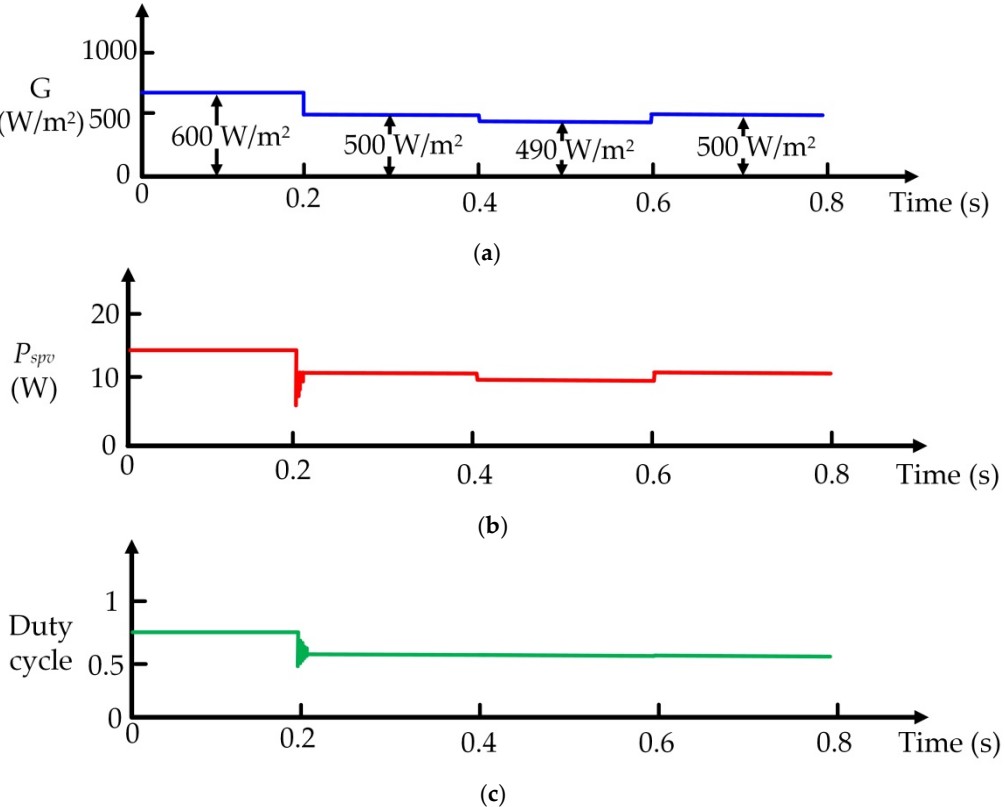

**Figure 4.** Corresponding SPV module for (**a**) solar radiation, G, (W/m²), (**b**) SPV module output power, $P_{spv}$, (W) and (**c**) duty cycle waveforms.

The change in the $R_{spv}$ of the SPV module is an important factor for MPPT. This proposed algorithm not only detects the SPV Module $P_{spv}$-$V_{spv}$ characteristic curves but also utilizes the CTC based on $R_{spv}$ to track the MPP. Furthermore, the proposed algorithm is suitable for poor climates (e.g., rain, cloud and shadow).

Figure 5 shows the proposed algorithm flowchart. First, the proposed algorithm implements the P&O algorithm, then reaches MPP; the duty cycle is fixed. Secondly, the proposed algorithm enters the CTC threshold control where $dV_{spv}(n) = V_{spv}(n) - V_{spv}(n-1)$; $dP_{spv}(n) = P_{spv}(n) - P_{spv}(n-1)$; the present SPV module output current is $I_{spv}$; the present solar radiation is G; dG = $|G(n) - G(n-1)|$; *a*, *b* and *c* are the parameters of Equations (3) and (4) and *d*, *e* and *f* are the parameters of Equation (5).

Figure 6 illustrates the boost converter with the MPPT algorithm-embedded diagram [26]. Its main elements include an inductor (*L* of 1 mH), a power MOSFET ($S_1$), a diode (*D*) and a capacitor ($C_{out}$ of 220 μF). It includes feedback circuits of an optical coupler and a current transducer. Further, it detects the $V_{spv}$ and $I_{spv}$ and transmits the signals to the microcontroller unit (MCU). The MCU (Microchip Technology, model number 18F452) outputs the PWM signal (PWM frequency of 30 kHz) and then drives the gate driver to control $S_1$ and reach the MPP.

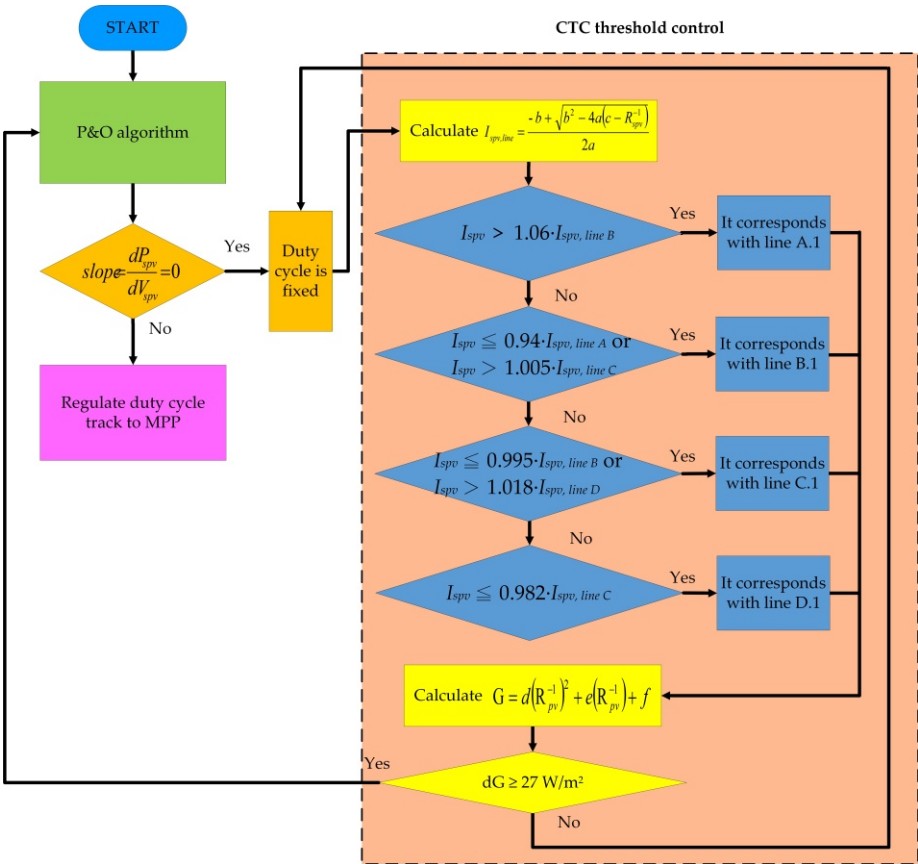

**Figure 5.** The proposed algorithm flowchart.

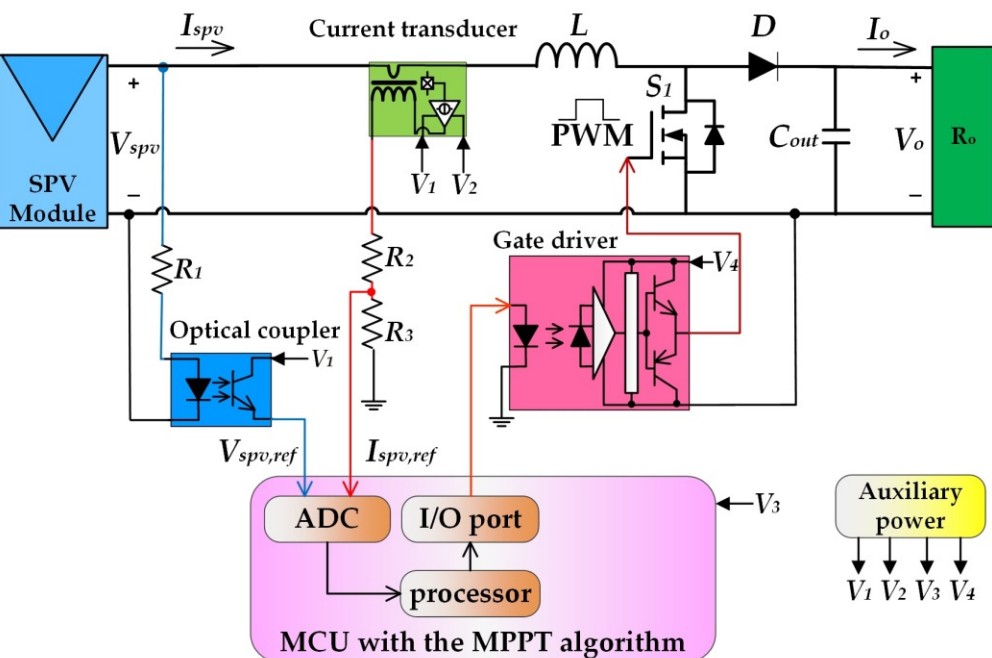

**Figure 6.** The boost converter with the MPPT algorithm-embedded diagram.

### 4. Experimental Results

Figure 7 shows the experimental SPV module and the prototype setup. The SPV module (model number Q025, Everbright, Beijing, China) G of 1000 W/m$^2$ and T of 25 °C specifications are as follows: $V_{MPP}$ = 8.3 V, $I_{MPP}$ = 2.4 A and $P_{MPP}$ = 20 W. In this experiment, the SPV module output power was connected to the input of the boost converter and the boost converter output was connected to the load. The MCU was employed to perform the MPPT control. The MCU outputted the PWM signal to drive the boost converter power MOSFET, $S_1$, which then reached the MPP.

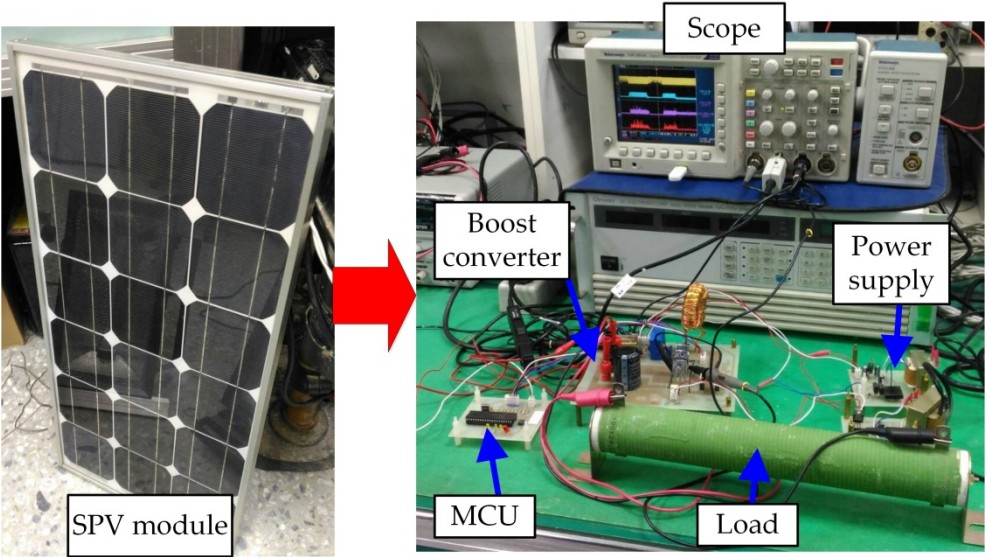

**Figure 7.** The experimental SPV module and the prototype setup.

In order to verify the proposed algorithm and the P&O algorithm performance, this study ran an experimental test under varying solar radiation and partial shading conditions. The results of the experimental proposed algorithm performance were higher than the P&O algorithm (Figures 8 and 9).

Figure 8 displays the comparison between the proposed algorithm and the P&O algorithm test results when the varying irradiance of 500 W/m$^2$ dropped to 220 W/m$^2$ then increased to 500 W/m$^2$ with a T of 25 °C. Figure 8a shows that the proposed algorithm's MPPT was activated. When time = $t_0$, the SPV module $R_{spv}^{-1}$ = 0.154 S, $V_{spv}$ = 8.5 V, $I_{spv}$ = 1.32 A and $P_{spv}$ = 11.22 W. According to Equation (4), the following were calculated: $I_{spv,lineA}$, $I_{spv,lineB}$, $I_{spv,lineC}$ and $I_{spv,lineD}$, respectively. The $I_{spv,lineC}$ = 1.299 A and $I_{spv}$ > 1.005·$I_{spv, lineC}$. Thus, the $I_{spv}$ fell on line B (Figure 2), which corresponded with line B.1 (Figure 3). Equation (5) was used to calculate the G = 500 W/m$^2$. At time = $t_1$, the SPV module $R_{spv}^{-1}$ = 0.072 S, $V_{spv}$ = 8.5 V, $I_{spv}$ = 0.62 A and $P_{spv}$ = 5.3 W. According to Equation (4), the following were calculated: $I_{spv,lineA}$, $I_{spv,lineB}$, $I_{spv,lineC}$ and $I_{spv,lineD}$, respectively. The $I_{spv,lineC}$ = 0.6 A and $I_{spv}$ > 1.005·$I_{spv,lineC}$. Therefore, the $I_{spv}$ fell on line B (Figure 2), which corresponded with line B.1 (Figure 3). Equation (5) was used to calculate the G = 220 W/m$^2$. At time = $t_2$, the SPV module $R_{spv}^{-1}$ = 0.154 S, $V_{spv}$ = 8.5 V, $I_{spv}$ = 1.32 A and $P_{spv}$ = 11.22 W. According to Equation (4), the following were calculated: $I_{spv,lineA}$, $I_{spv,lineB}$, $I_{spv,lineC}$ and $I_{spv,lineD}$, respectively. The $I_{spv,lineC}$ = 1.299 A and $I_{spv}$ > 1.005·$I_{spv,lineC}$. Thus, the $I_{spv}$ fell on line B (Figure 2, which corresponded with line B.1 (Figure 3). Equation (5) was used to calculate the G = 500 W/m$^2$. The proposed algorithm could accurately calculate the G and adjust the duty cycle track to the MPP. When the G was constant, the duty cycle was fixed. Therefore, the proposed algorithm caught the MPP accurately.

Figure 8b displays the P&O algorithm test results. This algorithm's MPPT was activated and perturbed to track the MPP and then the perturb method resulted in power loss. When the P&O algorithm at time = $t_0$ and a G of 500 W/m$^2$, at time = $t_1$, the G of 500 W/m$^2$ dropped to 220 W/m$^2$ and at time = $t_2$, the G of 220 W/m$^2$ rose to 500 W/m$^2$. The experiment results verified that the proposed algorithm's MPPT efficiency was better than the P&O algorithm (as in Table 1).

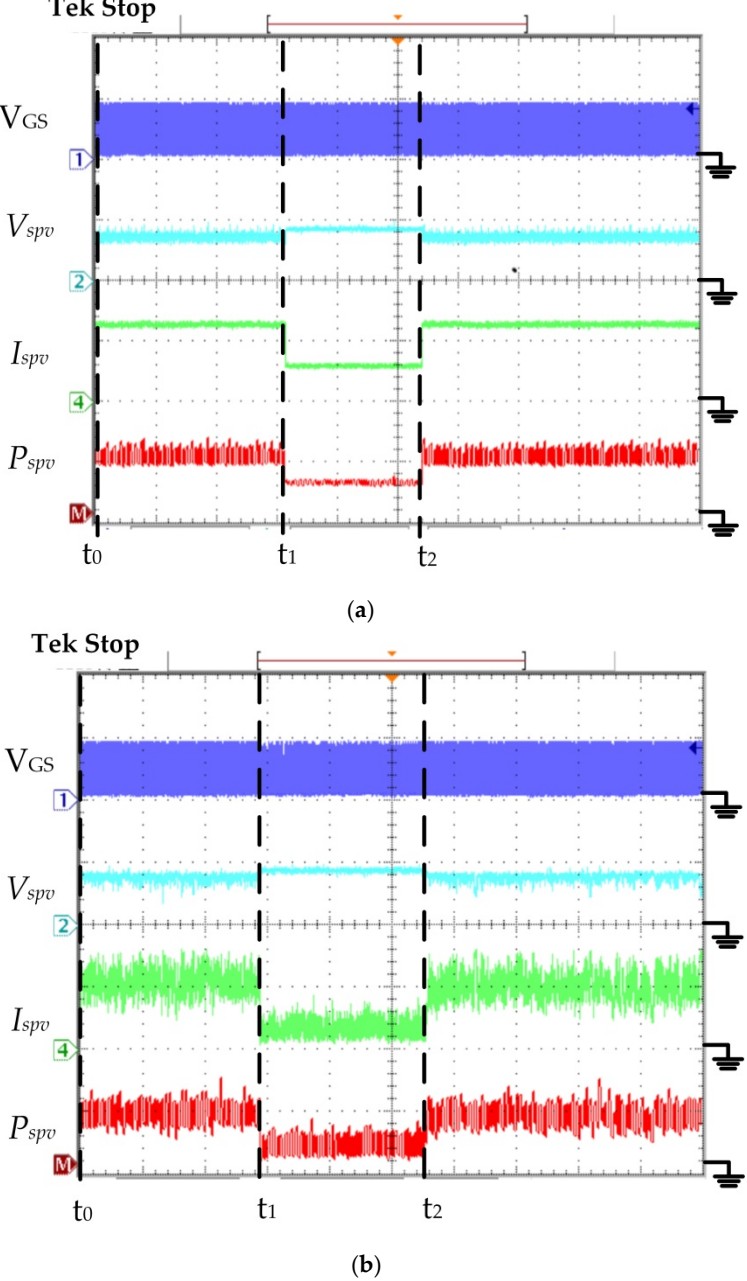

(**a**)

(**b**)

**Figure 8. Figure 8.** $V_{GS}$, $V_{spv}$, $I_{spv}$ and $P_{spv}$ waveforms for a SPV module under a T of 25 °C and varying irradiance of 500 W/m$^2$ to 220 W/m$^2$ then to 500 W/m$^2$: (**a**) the proposed algorithm and (**b**) the P&O algorithm. ($V_{GS}$: 20 V/div; $V_{spv}$: 10 V/div; $I_{spv}$: 1 A/div; $P_{spv}$: 10 W/div; Hor: 4 s/div).

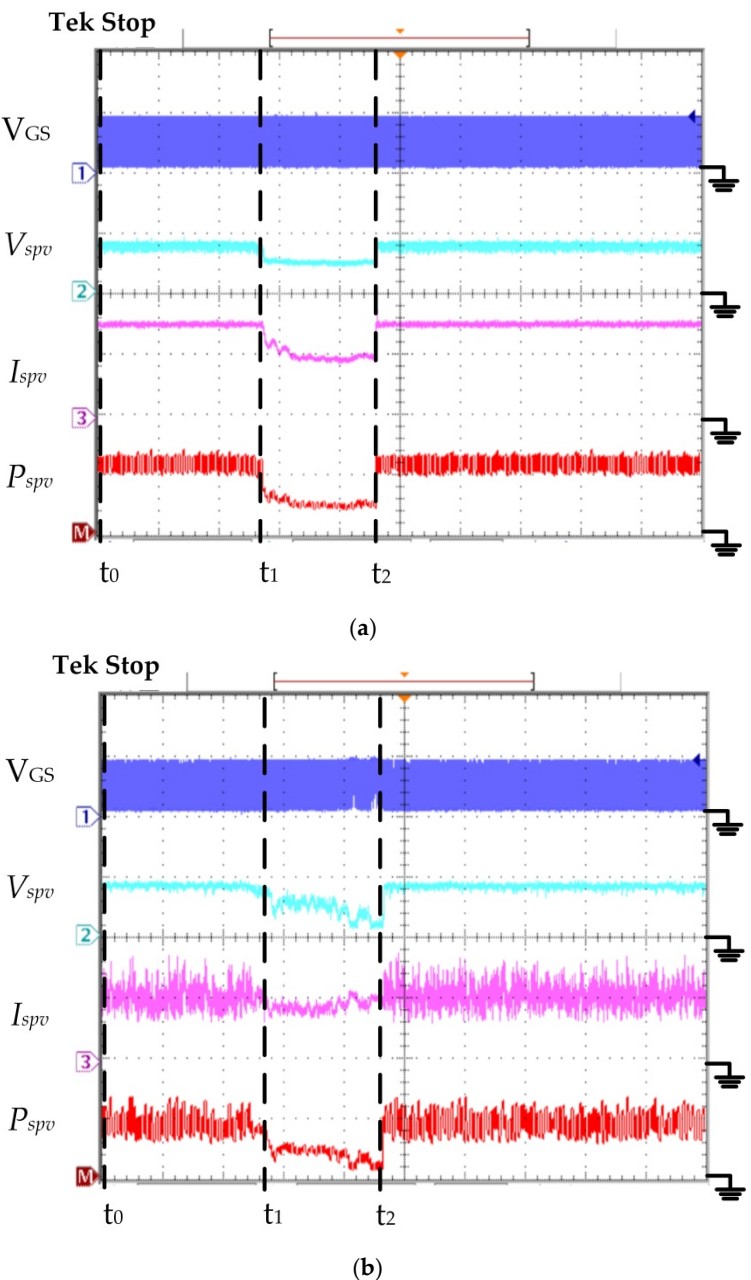

**Figure 9.** $V_{GS}$, $V_{spv}$, $I_{spv}$ and $P_{spv}$ waveforms for a SPV module under a T of 25 °C and partial shading conditions: (**a**) the proposed algorithm and (**b**) the P&O algorithm. ($V_{GS}$: 20 V/div; $V_{spv}$: 10 V/div; $I_{spv}$: 1 A/div; $P_{spv}$: 10 W/div; Hor: 4 s/div).

**Table 1.** Comparison efficiency of the proposed and P&O algorithm under various solar radiation and partial shading conditions.

| Algorithm | Various Solar Radiation | | Partial Shading Conditions |
|---|---|---|---|
| | G of 500 W/m² Drop to 220 W/m² | G of 220 W/m² Rise to 500 W/m² | |
| Proposed | 99% | 99% | 99% |
| P&O | 96% | 96% | 80% |

Figure 9 shows the comparison between the proposed algorithm and the P&O algorithm test results in partial shading conditions. The G and T were respectively 540 W/m$^2$ and 25 °C. Figure 9a shows that the proposed algorithm MPPT was activated. When the proposed algorithm at time = $t_0$, the SPV module $R_{spv}^{-1}$ = 0.166 S, $V_{spv}$ = 9 V, $I_{spv}$ = 1.5 A and $P_{spv}$ = 13.5 W. According to Equation (4), the following were calculated: $I_{spv,lineA}$, $I_{spv,lineB}$, $I_{spv,lineC}$ and $I_{spv,lineD}$, respectively. The $I_{spv,lineC}$ = 1.41 A and $I_{spv} > 1.005 \cdot I_{spv,lineC}$. Therefore, $I_{spv}$ fell on line B (Figure 2), which corresponded with line B.1 (Figure 3). Equation (5) was used to calculate the G = 540 W/m$^2$. At time = $t_1$, the SPV module suffered 1/2 partial shading conditions $P_{spv}$ = 6 W. The proposed algorithm was provided by the P&O algorithm with a quick response and accurately calculated the G and adjusted the duty cycle track to the MPP. When the G was constant, the duty cycle was fixed. Therefore, the proposed algorithm stably caught the MPP. At time = $t_2$, the SPV module $R_{spv}^{-1}$ = 0.166 S, $V_{spv}$ = 9 V, $I_{spv}$ = 1.5 A and $P_{spv}$ = 13.5 W. According to Equation (4), the following were calculated: $I_{spv,lineA}$, $I_{spv,lineB}$, $I_{spv,lineC}$ and $I_{spv,lineD}$, respectively. The $I_{spv,lineC}$ = 1.41 A and $I_{spv} > 1.005 \cdot I_{spv,lineC}$, so $I_{spv}$ fell on line B (Figure 2), which corresponded with line B.1 (Figure 3). Equation (5) was used to calculate the G = 540 W/m$^2$. Similarly, the proposed algorithm caught the MPP accurately.

Figure 9b shows the P&O algorithm test results. This algorithm's MPPT was activated and perturbed to track the MPP. However, the perturb method could converge to the LPPP, resulting in a lower system efficiency. When the P&O algorithm at time = $t_0$ and G of 540 W/m$^2$, at time = $t_1$, the SPV module suffered 1/2 partial shading conditions $P_{spv}$ = 4.7 W and at time = $t_2$, a G of 540 W/m$^2$. Similarly, the experiment results verified that the proposed algorithm's MPPT efficiency was higher than the P&O algorithm (as in Table 1).

## 5. Conclusions

The proposed algorithm by the P&O algorithm was combined with the solar radiation value detection scheme where the solar radiation value detection was based on the SPV module equivalent CTC. Further, it could operate consistently at the MPP under varying solar radiation and partial shading conditions. Therefore, this proposed algorithm could improve the P&O algorithm perturbation problem, avoiding actuating point oscillations near the MPP. Furthermore, this algorithm could converge to the MPP under partial shading conditions. The experiment results showed that the proposed algorithm under varying solar radiation (500 W/m$^2$ to 220 W/m$^2$ then to 500 W/m$^2$) and partial shading conditions reached 99% of MPPT efficiency. Thus, the proposed algorithm was better than the P&O algorithm. Accordingly, the proposed algorithm was confirmed to be of a high performance under various solar radiation and partial shading conditions.

In this research, the SPV system focused on the MPPT algorithm. This system can already provide maximum power to provide load under varying solar radiation and partial shadow conditions. However, it is well-known that SPV systems cannot provide power at night and must be connected to the grid, batteries, wind power and hydropower to provide users with sufficient power quality. Therefore, the system's operation matching between SPV systems and other power equipment is an important topic for future research.

**Author Contributions:** A.-S.J., H.-D.L. and S.-D.L. conceived the presented idea, designed, experimented and wrote this article; C.-H.L. supervised the findings of this work; all authors provided critical feedback and helped shape the research, analysis and manuscript. All authors have read and agreed to the published version of the manuscript.

**Funding:** This research was funded by the Ministry of Science and Technology, Taiwan, R.O.C., grant number MOST 110-2221-E-011-081.

**Institutional Review Board Statement:** Not applicable.

**Informed Consent Statement:** Not applicable.

**Acknowledgments:** The authors sincerely appreciate much support from the Taiwan Building Technology Center from The Featured Areas Research Center Program within the framework of the Higher Education Sprout Project by the Ministry of Education in Taiwan.

**Conflicts of Interest:** The authors declare no conflict of interest.

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
