# Peer review of "New Control Scheme for Solar Power Systems under Varying Solar Radiation and Partial Shading Conditions"

_processes, doi:10.3390/pr9081359_

Round 1

Reviewer 1 Report

Title: New Control Scheme for Solar Power Systems under Varying Solar Radiation and Partial Shading Conditions

Revision:

The paper wants to present a new control scheme for Photovoltaic unit connected to Power System. The proposed control algorithm deals with manage properly rapid irradiance variation and partial shading conditions, proposing improved performances compare to the conventional P&O MPPT algorithm. The paper is really interesting, but some points on which authors must work, must be highlighted:

  1. The template is not correct. You use Times New Roman, but the correct one is Palatino Linotype. Please correct.
  2. Please check the English in the introduction. In particular in some sentences the verb is missing (line 27 when you define the low radiation and lines 48 49 50).
  3. I deeply suggest citing these two works in order to improve the quality of paper bibliographic research of the introduction:
  • Rosini, A., Mestriner, D., Labella, A., Bonfiglio, A., & Procopio, R. (2021). A decentralized approach for frequency and voltage regulation in islanded PV-Storage microgrids. Electric Power Systems Research193, 106974.
  • Palmieri, A., Rosini, A., Procopio, R., & Bonfiglio, A. (2020). An MPC-Sliding Mode Cascaded Control Architecture for PV Grid-Feeding Inverters. Energies13(9), 2326.

These two papers show the application of new control techniques to Photovoltaic unit, and more precisely discuss the application of innovative controller as Model Predictive Control and Sliding Mode which need to be considered in this paper because these are the actuators of the references coming from the MPPT algorithm, and a good MPPT algorithm must have robust and reliable actuators (i.e. current regulators) in order to guarantee a perfect tracking. Please cite these two papers.

  1. Please clarify your algorithm. Cause to the poor English quality, your algorithm explanation is not easy to understand.
  2. Please improve the quality of the figures, inserting label and units of measure.

Author Response

Reply to Reviewer 1

The paper wants to present a new control scheme for Photovoltaic unit connected to Power System. The proposed control algorithm deals with manage properly rapid irradiance variation and partial shading conditions, proposing improved performances compare to the conventional P&O MPPT algorithm. The paper is really interesting, but some points on which authors must work, must be highlighted:

1

The template is not correct. You use Times New Roman, but the correct one is Palatino Linotype. Please correct.

Reply:

Many thanks to the Reviewer’s reminding. We have corrected the template font as the Palatino Linotype.

2

Please check the English in the introduction. In particular in some sentences the verb is missing (line 27 when you define the low radiation and lines 48 49 50).

Reply:

Many thanks to the Reviewer’s valuable suggestions. We have revised the sentences for this part.

3

I deeply suggest citing these two works in order to improve the quality of paper bibliographic research of the introduction:

•    Rosini, A., Mestriner, D., Labella, A., Bonfiglio, A., & Procopio, R. (2021). A decentralized approach for frequency and voltage regulation in islanded PV-Storage microgrids. Electric Power Systems Research, 193, 106974.

•    Palmieri, A., Rosini, A., Procopio, R., & Bonfiglio, A. (2020). An MPC-Sliding Mode Cascaded Control Architecture for PV Grid-Feeding Inverters. Energies, 13(9), 2326.

These two papers show the application of new control techniques to Photovoltaic unit, and more precisely discuss the application of innovative controller as Model Predictive Control and Sliding Mode which need to be considered in this paper because these are the actuators of the references coming from the MPPT algorithm, and a good MPPT algorithm must have robust and reliable actuators (i.e. current regulators) in order to guarantee a perfect tracking. Please cite these two papers.

Reply:

Many thanks to the Reviewer’s reminding. We have added these two articles in references [2] and [3].

4

Please clarify your algorithm. Cause to the poor English quality, your algorithm explanation is not easy to understand.

Many thanks to the Reviewer’s valuable suggestions. We have English proofreading by a native speaker.

5

Please improve the quality of the figures, inserting label and units of measure.

Many thanks to the Reviewer’s valuable suggestions. We have updated all of the figures, inserting labels and units of measure in this manuscript, and improved the image quality.

Reviewer 2 Report

The authors present a very interesting work for the scientific community. However, the focus is very poor and the quality of the figures and the writing leaves much to be desired, several corrections must be made for approval.
The abstract should be expanded a bit by including some of the most relevant results obtained in the study.
The introduction should be expanded, referring to previous works and describing them. The authors concentrate 19 references in 17 introductory lines, they must be developed.
Do not use Fig. 1, use Figure 1. Do this with all figures. In addition, Figure 1 is well known, highlighting the relevance of this research.
Use the International System of Units and its abbreviations. i.e. Line 62 Coulombs = C.
Figure 2, put the images in parallel and enlarge the size. In addition, the figures should be better discussed and their most relevant results described.
In general, the description of all the figures is very poor, no legends are used in the figures and the text below the graphic is used. Also, some images, such as Figure 6, are of very low quality and some words are unreadable.
Better describe Figure 8.
Figures 9 and 10 are of very low quality, the values ​​can hardly be appreciated in the represented graphs.
The conclusions need to be expanded significantly.
The references do not contain DOI of the articles.

Author Response

Reply to Reviewer 2

The authors present a very interesting work for the scientific community. However, the focus is very poor and the quality of the figures and the writing leaves much to be desired, several corrections must be made for approval.

1

The abstract should be expanded a bit by including some of the most relevant results obtained in the study. The introduction should be expanded, referring to previous works and describing them. The authors concentrate 19 references in 17 introductory lines, they must be developed.

Reply:

Many thanks to the Reviewer’s reminding.

1. We have added the relevant actual test results to the abstract, and further explain.

2. We have rewritten the introduction and cited more references [4-23] and introduced the existing technology more clearly.

2

Do not use Fig. 1, use Figure 1. Do this with all figures. In addition, Figure 1 is well known, highlighting the relevance of this research.

Reply:

Many thanks to the Reviewer’s valuable suggestions. We have deleted figure 1 and related descriptions.

3

Use the International System of Units and its abbreviations. i.e. Line 62 Coulombs = C.

Reply:

Many thanks to the Reviewer’s reminding. We have deleted figure 1 and also delete this related description.

4

Figure 2, put the images in parallel and enlarge the size. In addition, the figures should be better discussed and their most relevant results described.

Many thanks to the Reviewer’s valuable suggestions. We have put the figure 2 (changed to figure 1) images in parallel and enlarge the size. And further describe the figure content.

5

In general, the description of all the figures is very poor, no legends are used in the figures and the text below the graphic is used. Also, some images, such as Figure 6, are of very low quality and some words are unreadable.

Many thanks to the Reviewer’s reminding. We have updated all of the figures in this manuscript and improved the image quality.

6

Better describe Figure 8.

Many thanks to the Reviewer’s valuable suggestions. We have rewritten the description of figure 8 (changed to figure 7).

7

Figures 9 and 10 are of very low quality, the values can hardly be appreciated in the represented graphs.

Many thanks to the Reviewer’s reminding. We have updated figures 9 and 10 (changed to figure 8 and 9) in this manuscript and improved the image quality.

8

The conclusions need to be expanded significantly.

Many thanks to the Reviewer’s reminding. We have expanded significantly in the conclusions, as follows

In this research, SPV systems focused on the MPPT algorithm. This system can al-ready provide maximum power to provide load under varying solar radiation and par-tial shadow conditions. However, it is well known that SPV systems cannot provide power at night and must be connected to the grid, batteries, wind power, and hydro-power to provide users with sufficient power quality. Therefore, the system’s operation matching between SPV systems and other power equipment is an important topic for future research.

9

The references do not contain DOI of the articles.

Many thanks to the Reviewer’s valuable suggestions. We have added DOI in the references of the articles.

Round 2

Reviewer 2 Report

The authors have made all the proposed changes